# Anti-Inflammatory Potential of Cow, Donkey and Goat Milk Extracellular Vesicles as Revealed by Metabolomic Profile

**DOI:** 10.3390/nu12102908

**Published:** 2020-09-23

**Authors:** Samanta Mecocci, Federica Gevi, Daniele Pietrucci, Luca Cavinato, Francesco R. Luly, Luisa Pascucci, Stefano Petrini, Fiorentina Ascenzioni, Lello Zolla, Giovanni Chillemi, Katia Cappelli

**Affiliations:** 1Dipartimento di Medicina Veterinaria, University of Perugia, 06123 Perugia, Italy; samanta.mecocci@studenti.unipg.it (S.M.); luisa.pascucci@unipg.it (L.P.); 2Centro di Ricerca sul Cavallo Sportivo, University of Perugia, 06123 Perugia, Italy; 3Dipartimento di Scienze Ecologiche e Biologiche, Università della Tuscia, 01100 Viterbo, Italy; gevi@unitus.it (F.G.); zolla@unitus.it (L.Z.); 4Dipartimento per l’Innovazione Nei Sistemi Biologici, Agroalimentari e Forestali, Università della Tuscia, 01100 Viterbo, Italy; daniele.pietrucci.89@gmail.com; 5Dipartimento di Biologia e Biotecnologie C. Darwin, Università di Roma la Sapienza, 00185 Roma, Italy; luca.cavinato@uniroma1.it (L.C.); francesco.luly@uniroma1.it (F.R.L.); fiorentina.ascenzioni@uniroma1.it (F.A.); 6Istituto Zooprofilattico Sperimentale dell’Umbria e delle Marche, 06126 Perugia, Italy; s.petrini@izsum.it; 7Institute of Biomembranes, Bioenergetics and Molecular Biotechnologies, IBIOM, CNR, 70126 Bari, Italy

**Keywords:** Milk Extracellular Vesicles (MEVs), metabolomics, immunomodulation, inflammation, mass spectroscopy

## Abstract

In recent years, extracellular vesicles (EVs), cell-derived micro and nano-sized structures enclosed in a double-layer membrane, have been in the spotlight for their high potential in diagnostic and therapeutic applications. Indeed, they act as signal mediators between cells and/or tissues through different mechanisms involving their complex cargo and exert a number of biological effects depending upon EVs subtype and cell source. Being produced by almost all cell types, they are found in every biological fluid including milk. Milk EVs (MEVs) can enter the intestinal cells by endocytosis and protect their labile cargos against harsh conditions in the intestinal tract. In this study, we performed a metabolomic analysis of MEVs, from three different species (i.e., bovine, goat and donkey) by mass spectroscopy (MS) coupled with Ultrahigh-performance liquid chromatography (UHPLC). Metabolites, both common or specific of a species, were identified and enriched metabolic pathways were investigated, with the final aim to evaluate their anti-inflammatory and immunomodulatory properties in view of prospective applications as a nutraceutical in inflammatory conditions. In particular, metabolites transported by MEVs are involved in common pathways among the three species. These metabolites, such as arginine, asparagine, glutathione and lysine, show immunomodulating effects. Moreover, MEVs in goat milk showed a greater number of enriched metabolic pathways as compared to the other kinds of milk.

## 1. Introduction

Newborn mammals rely on milk as the main source of nutrition. Milk is not just food but it represents a sophisticated signaling system that delivers maternal milk-derived messages to promote postnatal health. Milk was found to play an important role in the development of newborn immune system, based on evidence that the composition of milk can modulate the immune response [1].

In recent years, studies on the molecular mechanisms underlying this mother to child information transfer have increased. Among them, immune modulatory, anti-infective and anti-inflammatory properties of milk emerged and seemed to be mediated by signaling molecules enclosed in micro and nano-sized membrane-bound structures called Extracellular Vesicles (EVs) and capable of carrying several types of information [2,3].

EVs are generally classified into two large categories as Microvesicles (MVs) and Exosomes. MVs range in size from 100 to 2000 nm and are delivered through the outward budding of the plasma membrane. EXs are 30–100 nm-sized vesicles generated from the endosomal compartment through the inward budding of the outer membrane of multivesicular bodies (MVBs) and their fusion with the plasma membrane.

In this paper we choose to refer to EVs as a whole, indicating all the vesicles we isolated from milk without discrimination of one specific type, due to conflicting definitions related to their function, dimension or cell origin [4,5,6].

EVs have the key features for intercellular communication and this could allow their use in different clinical applications. Among them, high specificity for the target, great stability and ability to cross biological barriers are well documented [7]. EVs of different origins can induce many various responses in the target tissue, for example, enhancing tissue regeneration [8,9] or participating in immune modulation and modification of inflammatory conditions [10,11,12,13] and could represent a future alternative to stem cell therapies in some fields [14,15].

EVs can explicate their biological function near the production site with a paracrine action but can also act remotely (endocrine activity), inducing the reprogramming of distant target cells [16,17,18,19].

Although EVs are released in the extracellular environment by virtually all cell types and have been recovered from every biological fluid, including blood, urine, bronchoalveolar lavage fluid, saliva and bile [20,21,22,23], milk is one of the most promising scalable sources of EVs [24].

Milk EVs (MEVs) were first described in 1971. However, they were identified as such only in 2007 by Admyre et al. [24], who highlighted the immunological properties of EVs collected from human colostrum and mature milk. MEVs exhibit some key features as the ability to enter the intestinal and endothelial cells by endocytosis and to protect labile cargos against harsh conditions in the intestinal tract [25,26,27]. Once absorbed, they can flow into the bloodstream, modifying the gene expression of cells even very distant [28].

It has been observed that MEVs present in human and bovine milk can enter fibroblasts, macrophages and vascular endothelial cells [27,29,30]. MEVs from pig milk and the miRNAs contained in them are also taken up by intestinal epithelial cells promoting their proliferation [31]. It has also been shown that MEVs can enter cultured cells, including epithelial cells, monocytes/macrophages and dendritic cells [25,32]. More recently, the first study on the biodistribution of bovine MEVs in mouse and porcine models has been published [33].

The set of these experimental evidences suggests a possible role of MEVs in the inflammatory and immune response.

Immunomodulatory activity of milk has been proven for humans, cows and donkeys [34]. In the latter species, it seems to have effects similar to human milk and higher than the bovine one. In particular, it has been speculated that donkey and human milk have anti-inflammatory properties, modulating pro and anti-inflammatory cytokine balance [34,35,36].

While the majority of previous EVs studies deal with protein and nucleic acid cargos, the role of other components such as metabolites has been overlooked until recently [29,37,38,39,40,41]. Metabolites represent a bio-functional category, also composed of low-molecular-weight (<1 kDa) components, that generally refer to small end product biomolecules or intermediates involved in metabolic processes such as alcohols, amides, amino acids, carboxylic acids and sugars [42].

Metabolomics approaches, such as mass spectrometry (MS)-based analysis, enable the identification and quantitation of a huge variety of small molecules and are widely used to profile the metabolome [43]. The entire metabolites content of EVs that is constituted by organic acids, amino acids, sugars and their conjugates, nucleotides and nucleosides, cyclic alcohols, carnitines, aromatic compounds and vitamins have been described, often focusing on a possible role of metabolites as cancer biomarkers from urine and blood [44,45]. At the best of our knowledge, there are no studies on MEV metabolites as immunomodulators and anti-inflammatories, while these properties have been observed in EVs [11].

Unfortunately, MEVs share several physicochemical properties with other nanostructures present in milk, such as fat globules, casein micelles and aggregated proteins. This homogeneity makes MEVs isolation a challenging task that, in this study, we tried to overcome through separation on a sucrose gradient. Basing on the actual aforementioned knowledge on EVs features as well as the little known MEVs, we adopted an untargeted approach using ultrahigh performance liquid chromatography (UHPLC)–Orbitrap-MS, to characterize MEV metabolomes derived from the milk obtained by bovine, goat and donkey. Indeed, the aim of the study was to expand our knowledge on these particular structures produced by cells and poured into milk, evaluating the information carried by the metabolome MEVs cargo and the variability between the three species. Metabolites significantly enriched in the MEV fractions with respect to all the others were identified, both commons to all kinds of milk or peculiar of a species and a pathway enrichment analysis was carried out to highlight the biological relevance of the metabolic pathways and to hypothesize probable biological functions (Figure 1).

## 2. Materials and Methods

### 2.1. Milk Collection

Samples were collected from mass milk for each species (cow, goat, donkey), to avoid individual variability. For each milk, four different samples were analyzed. Milk was stored at 4 °C for less than 24 h before processing to separate whey fraction (richer in MEVs than the fat one) without any intermediate cryo-preservation to minimize artefacts. Milk was taken from farms, located in Umbria (Italy), selected and monitored by the Veterinary Medicine Department, University of Perugia—two local farms for goat and donkey and the Didactic Zootechnical Farm of University for bovine.

### 2.2. EVs Isolation

MEVs were separated by differential centrifugation (DC) following a protocol developed for human milk [46] and subsequent fractionation on sucrose gradients. Additionally, goat milk was treated with ethylenediaminetetraacetic acid tetrasodium salt dihydrate (EDTA) as reported by Vaswani and collaborators [47]. Common preliminary centrifugation steps were used to eliminate cellular debris and protein complexes in the pellet and fat globules on the surface, recuperating the supernatant and leaving about 5 mL in the pellet and 5 mL in the upper phase. Starting from 180 mL for cow and donkey milk, two consecutive 3000× *g* centrifugations for 10 min at room temperature followed by a 5000× *g* and a 10,000× *g* centrifugations for 30 min at 4 °C were executed. For the 180 mL of goat milk, after the two 3000× *g* initial centrifugations, 0.25 M EDTA (pH 7,4) was added to the supernatant in equal volume (1:1) and, after an incubation for 15 min on ice, samples were centrifuged at 5000× *g* for 30 min at 4 °C (3000× *g* and 5000× *g* centrifugations were performed in a Eppendorf^®^ Centrifuge 5810R with a F34-6-38 rotor, while a Beckman Coulter Optima L-100 XP with an SW41 Ti rotor was used for centrifugations from 10,000× *g* to 192,000× *g*). Then a series of centrifugations at 10,000× *g*, a 35,000× *g* and a 70,000× *g* each for 60 min and at 4 °C were carried out using polyallomer tubes. From here on out, all the steps continued in common for both procedures. A sucrose gradient in polyallomer tubes composed by 350 μL per fifteen fractions from 2.0 M to 0.4 M loaded on a 700 μL of sucrose 2,5 M was prepared. The 10,000× *g* supernatant obtained from the first procedure and the same quantity of 70,000× *g* supernatant for the second one were loaded on the sucrose and were kept at 192,000× *g* centrifugation overnight (about 18 h) at 4 °C. After that, phases of 460 μL from the bottom to the top of the tube were recovered and merged together going to constitute five pools of fractions as follows—pool 1 (P1, pellet + 1 to 5 fractions), pool 2 (P2, from 6 to 9), pool 3 (P3, from 10 to 13), pool 4 (P4, from 14 to 18) and pool 5 (P5, from 19 to 24 fractions). Afterward, a PBS lavage of each pool was made recovering the vesicles in the pellet through a final ultracentrifugation at 100,000× *g* for 1 h at 4 °C and preserving them at −80 °C until next usage. All details are summarized in Figure 1a. As reported in Zonnevelt et al. [46], MEVs should be recovered in pellets derived from 6 to 9 fractions that correspond to the P2 pool of our study.

### 2.3. MEVs Characterization

#### 2.3.1. Western Blotting

Protein extraction was performed in 20–40 μL, of lysis buffer (4% SDS, 100 mM Tris-HCl and 100 mM DTT); followed incubation at 95 °C for 5 min, the lysates were centrifuge at 17,000× *g* for 20 min at 4 °C. Protein quantification was performed with the Bradford assay (Merck KGaA, Darmstadt, Germany). Protein samples of 25–30 μg were analyzed by immunoblotting on 10% SDS-PAGE as previously described [48]. The primary antibodies were—CD81 (bs-6934R, Bioss Antibodies, Woburn, MA, USA), diluted 1:500; TSG-101 (sc-7964, Santa Cruz, CA, USA), diluted 1:400; Calnexin (sc-23954, Santa Cruz, CA, USA), diluted 1:400. Membranes were visualized with a Chemi Doc XRS system (Bio-Rad Laboratories Ltd., Hemel Hempstead, UK) and images processed with ImageLab (BioRad, Hercules, CA, USA).

#### 2.3.2. Transmission Electron Microscopy (TEM)

Approximately 10 μL of MEV suspension from each species, were placed on Parafilm. A Formvar-coated copper grid (Electron Microscopy Sciences, Hatfield, PA, USA) was gently placed on the top of each drop for about 20 min in a humidified chamber. Grids were then washed in PBS and distilled water. Finally, they were contrasted with 2% uranyl acetate for 5 min and air dried. The observation was performed using a Philips EM208 transmission electron microscope equipped with a digital camera (CUME—University Centre of Electron Microscopy, Perugia, Italy).

### 2.4. Metabolomic Analysis

#### 2.4.1. Metabolite Extraction

One mL of a solvent mixture previously stored at −20 °C composed by chloroform/methanol/water (1:3:1 ratio) was added to each sample. Then, samples were vortexed for 5 min and left on ice for 2 h for completed protein precipitation. The solutions were then centrifuged for 15 min at 15,000× *g* and were dried to obtain visible pellets. Finally, the dried samples were re-suspended in 0.1 mL of water, 5% formic acid and transferred to glass autosampler vials for LC/MS analysis.

#### 2.4.2. UHPLC-HRMS

Twenty-microliter of extracted supernatant samples was injected into an ultrahigh-performance liquid chromatography (UHPLC) system (Ultimate 3000, Thermo) and run on a positive mode—samples were loaded on to a Reprosil C18 column (2.0 mm × 150 mm, 2.5 μm-DrMaisch, Ammerbuch, Germany) for metabolite separation. Chromatographic separations were made at a column temperature of 30 °C and a flow rate of 0.2 mL/min. For positive ion mode (+) MS/MS analyses were performed. A linear gradient 0–100% of solvent A (ddH2O, 0.1% formic acid) to B (acetonitrile, 0.1%formic acid) was employed over 20 min, returning to 100% A in 2 min and holding solvent A for a 6-min post time hold. Acetonitrile, formic acid and HPLC-grade water and standards (≥98% chemical purity) were purchased from Sigma Aldrich. The UHPLC system was coupled online with a Q-Exactive Orbitrap mass spectrometer (Thermo, Waltham, MA, USA) scanning in full MS mode (2 μ scans) at resolution of 70,000 in the 67 to 1000 *m*/*z* range, a target of 1106 ions and a maximum ion injection time (IT) of 35 ms with 3.8 kV spray voltage, 40 sheath gas and 25 auxiliary gas. The system was operated in positive ion mode. Source ionization parameters were as follows—spray voltage, 3.8 kV; capillary temperature, 300 °C; and S-Lens level, 45. Calibration was performed before each analysis against positive ion mode calibration mixes (Pierce, Thermo Fisher, Rockford, IL, USA) to ensure error of the intact mass within the sub ppm range. Resolution MS2 was set at 17,500 (FWHM atm/z 200), Normalized Collision Energy (NCE) were set at 20. Metabolite assignments were performed using computer software (Maven, 18 Princeton, NJ, USA), upon conversion of raw files into a .mzXML format using MassMatrix (Cleveland, OH, USA). Analysis of each sample was performed in triplicate and *p*-value of 0.01 was used for all abundance comparisons between sets of triplicates.

### 2.5. Data Elaboration and Statistical Analysis

Technical replicates data were exported as .mzXML files and processed through MAVEN.4.0 software. Mass spectrometry chromatograms were created for peak alignment, matching and comparison of parent and fragment ions with tentative metabolite identification, within a 2 p.p.m. mass-deviation range between the observed and expected results against an imported Kyoto Encyclopedia of Genes and Genomes (KEGG) database. Statistical analysis of metabolomics data was performed in R vr. 3.5.3., using the MetaboAnalyst package vr. 2.0.1 [49]. Metabolomic data for each species (cow, goat, donkey) were analyzed independently, to detect key metabolites for each type of milk. Samples were grouped into five categories—P1, P2, P3, P4 and P5. Each category describes a different milk pool of sucrose gradient fractions, as discussed above (Figure 1a). Raw data (metabolite concentrations) were normalized using the normalization by sum and the Pareto scaling. Metabolite concentrations, which differ among fractions, were detected using the ANOVA test. Only metabolite with a False Discovery Rate less than 0.05 were considered, to take account for multiple testing correction. Significant metabolites were analyzed using the Fisher’s Least Significant Difference (LSD) test to explore the metabolite concentrations among fractions further. In detail, the Fisher’s LSD was used to identify only the metabolites which are highly abundant in the P2 pool containing MEVs (Figure 1b_1_). Normalized data for metabolites of interest were reported using boxplots with the ggplot2 package. To identify the most relevant metabolic pathways of differential metabolites (enriched in P2, Figure 1b_1_) and display their relationship with genes, proteins and other metabolites, the bioinformatic tool Metscape [50], a plugin for the open-source network data integration tool Cytoscape [51], was used allowing the construction of the Compound-Reaction-Enzyme-Gene Network (CREGN) (Figure 1c). Moreover, to assess the information carried by the whole metabolite cargo of MEVs (Figure 1b_2_), a Metabolite Set Enrichment Analysis (MSEA) by over-representation analysis (ORA) was performed using the MetaboAnalyst package [52] (Figure 1d).

## 3. Results

### 3.1. MEVs Characterization

#### 3.1.1. Western Blotting

Total protein content was evaluated on all the five pool of fractions recovered by gradient centrifugation and the most abundant ones were P2 and P3 for cow and P2–P4 for goat/donkey. In these pools the total amount of proteins was similar in cow and goat and about 5 times less in donkey, suggesting a lower protein content in donkey milk (Figure 2A).

Western blot (WB) analysis allowed us to identify the pools enriched with MEVs using the vesicle specific antigens CD81 and TSG101 and one vesicle-negative antigen, the calnexin [4], the latter being used to evaluate the presence/absence of cellular contaminants. The P2 pools appeared positive to both TSG101 and CD81 in all the species analyzed strongly suggesting that this pool is the most enriched in MEVs (Figure 2B). We observed a variation in the size of CD81 between MEVs and Jurkat cells but a similar size of CD81 in EV has been reported in the literature [5,53]. As clearly visible in Figure 2B, calnexin was detected only in the Jurkat lysates used as a positive control, strongly suggesting that the isolation procedure excluded other cellular organelles from the MEV enriched fractions. Collectively these data suggested that MEVs were more enriched in the P2 pool, which therefore was selected for further analysis.

#### 3.1.2. Transmission Electron Microscopy

At TEM, MEVs appeared to be mainly round in shape and ranged in size from 30 to 500 nm regardless of species. MEVs were observed as isolated vesicles or aggregated in clusters, showing a peripheral rim enclosing an electron-lucent to moderately electron dense content (Figure 3).

### 3.2. Metabolomic Analysis

Metabolomic data were collected for all the pools of the gradients (P1, P2, P3, P4, P5) and subsequently analyzed to identify the possible presence of metabolites enriched in the P2 pool which is the most enriched in MEVs as assessed by WB. For each species, explorative data analysis was performed using heatmaps (Appendix A). Subsequently, metabolites enriched in the P2 pool were identified using the ANOVA statistical tests. Normalized data of metabolites enriched in the P2 pool across all milks are reported in Appendix A.

The metabolome heatmap of the Bovine milk shows a clear clustering of the P2 pool (Appendix A), due to higher concentration of 17 metabolites (Figure 4, Appendix A), eleven of which (inosine, adenosine, adenine, guanosine, NADH, FMN, NAD, IMP, Nicotinamide ribotide, GMP, NADP) were highly enriched in the P2 pool compared to all the other pools while the remaining six (cytidine, cytosine, deoxyinosine, allantoin, lysine, pyridoxamine), although statistically enriched in the P2, were also present in a remarkable concentration in at least one remaining pool.

The metabolome of P2 pool in Donkey milk is mostly distinguishable (Appendix A). All samples from the P2 pools cluster together in the same group with the P1 pool, while P3, P4 and P5 grouped apart. Among the metabolites that cause the clustering of the P2 pools, 15 show a higher concentration in P2 (FDR < 0.05) (Figure 5, Appendix A). For nine of these metabolites (i.e., glutathione disulfide, NAD, AMP, dGMP, glutathione, dUTP, IMP, asparagine, diidothyronine), the levels in the P1, P3, P4 and P5 were very low, while for the other six (i.e., S-adenosyl-L-homocysteine, orotato, xanthosine, nicotinamide ribotide, adenosine, 2-dihydro-D-gluconate) a remarkable concentration in at least one remaining pool was observed.

Also, the metabolome of the Goat milk is characterized by specific metabolites in the P2 pool. The heatmap (Appendix A) shows that samples of the P2 grouped entirely apart from the P1, P3, P4 and P5 pools. The number of metabolites that are enriched in the P2 is equal to 27, higher than the number of metabolites found in Donkey and Bovine milk (Figure 6, Appendix A). Metabolites enriched in P2 pools in at least two milks are reported in Appendix A.

### 3.3. Network and Pathway Enrichment Analysis

To explain the biological meaning of the identified significantly enriched metabolites in MEVs, a network analysis between these molecules and genes, proteins and metabolites in related pathways was carried out. Four metabolic pathways were found common to all the species, even if in one case only one metabolite was involved and in particular “Purine”, “Pyrimidine”, “Urea cycle and metabolism of arginine, proline, glutamate, aspartate and asparagine” and “Vitamin B3” metabolisms (Table 1). Furthermore, donkey and goat had in common two more processes where their MEV metabolites were implicated—“methionine and cysteine” and “tyrosine” metabolism. The “Urea cycle and metabolism of arginine, proline, glutamate, aspartate and asparagine” network is reported in Figure 7 and Table 1 while all the other common networks for the three species are shown in Appendix A and Table 1.

Although the same metabolites and same pathways emerged from all types of milk, specific pathways have also been found for individual species. In particular, the goat MEVs showed a greater number of these metabolic pathways than the others and in particular “Glycine, serine, alanine and threonine”, “Tryptophan”, “Biopterin”, “Vitamin B9 (folate)”, “Glycolysis and Gluconeogenesis”, “Glycerophospholipid” and “Galactose” metabolisms. Bovine MEVs brought “Vitamin B2”, “B6” and “H” metabolic pathways in addition to “Lipoate” and “Lysine” metabolisms while donkey had the “Pentose phosphate” pathway as adjunctive information (Table 1). The networks of MEV metabolites and other compounds, genes, enzymes and reactions implicated in common pathways among species are visualized in Figure 7 and Appendix A, while Appendix A shows specific pathways for each kind of milk.

Since the aforementioned metabolites are only a part (enriched metabolites in P2—Figure 1b_1_) of those transported by the MEVs (whole P2—Figure 1b_2_) a Metabolite Set Enrichment Analysis (MSEA) on the entire vesicular content was performed and the results are summarized in Figure 8. Most of the pathways emerged from the CREGN analysis were reconfirmed in MSEA—metabolites proper to all the species like purine and pyrimidine metabolisms or arginine, aspartate and glutamate metabolism or cysteine and methionine metabolism and also metabolisms characteristic of a single species as in the case of thyroxine and tryptophan for goat and vitamin B6 for cow. At the same time, some specific pathways found for one species have been found for the others, for example, the metabolism of vitamin B6 which, in addition to the bovine, also emerged for goat or the metabolism of tryptophan initially found only in goat but here also in donkey. Furthermore, new pathways emerged from the MSEA of the whole MEV metabolome, such as those of aminoacyl-tRNA or taurine and hypotaurine biosynthesis for all the species, others such as valine, leucine and isoleucine biosynthesis were present in bovine and donkey or the phenylalanine metabolism in goat and donkey or again the metabolism of thiamine for cow and goat. A total Compound-Reaction-Enzyme-Gene Network (CREGN) for all the species is visible in the Appendix A.

## 4. Discussion

Milk is a very complex matrix, rich in a large number of different components, including EVs, micro/nanoparticles that have a key role in intercellular communication. EVs overall display some key features—the high specificity for the target, their natural composition in comparison to artificial nano-carriers, the great stability, the intrinsic ability to cross biological barriers [7]. These characteristics have made EVs suitable for using them as carriers for the administration of drugs, especially for the treatment of tumors or degenerative diseases and bioengineered EVs have been used for therapeutic applications. Indeed, studies on extracellular vesicles began precisely for these applications [54,55]. Among the functions that EVs induce in the target cells once taken up are known immunomodulating and anti-inflammatory properties [10,12,13,56], not yet demonstrated for the MEVs. It could be advantageous to evaluate if these properties also reside in MEVs since they have the capability to cross biological barriers and withstand adverse conditions as those of the gastrointestinal tract [25,26,27], maintaining the membrane integrity that allows protecting their cargo. Once arrived at the receiving cells, both intestinal and remote targets reached by entering the circulatory stream, MEVs interact with them releasing their content and fulfilling their action [28]. Moreover, we hypothesize that the ability of MEVs to transport all the active components found in them is a great advantage compared to free substances because it turns out to be like a package that supplies not one substance at a time but all together, maybe their strength. This could pave the way for possible future applications of MEVs to be used to improve conditions of chronic inflammation for example those of the gastrointestinal tract, with the advantage to use them separately from the whole milk to obtain an enriched product consequently more effective.

Bovine milk is undoubtedly the most used for human nutrition, although goat and donkey milks are also used in particular for their suitability to replace the human milk. To better characterize the molecular features of MEVs in the milk of these three species (cow, donkey and goat) the MEV metabolites cargo was evaluated. A preliminary Transmission Electron Microscopy and Western Blotting analysis have been applied to identify the gradient fractions containing the MEVs and their morphology and to evaluate the isolation method quality through the analysis of surface markers typical of EVs. According to the International Society for Extracellular Vesicles (ISEV) [4] guidelines and relevant bibliography [57], the pools were tested for CD81, TSG101 and Calnexin (Figure 2B) validating the procedure herein adopted for MEV isolation.

Then, we first tried to understand whether MEVs carry a peculiar enriched message that does not belong to the other portions of milk. For this purpose, we have identified the P2 pool as the most enriched in MEVs and statistically different metabolites of P2 with respect to the other milk pools were detected (Figure 1b_1_
Figure 4, Figure 5, and Figure 6, Appendix A). Moreover, in order to compare the MEVs of the three kinds of milk, any common or peculiar metabolite and pathway were highlighted (Figure 7 and Appendix A, Table 1). Then, the whole content of vesicle metabolites was evaluated through the identification of pathways involved in the entire MEV metabolite cargo that might induce additional responses in the recipient cell (Figure 1b_2_,d, Figure 8 and Appendix A).

From the statistical analysis on the differentially abundant metabolites in P2 compared to the other pools, the main typologies of components found in the MEVs were some amino acids/peptides, nucleotides/nucleosides and group B vitamins, very crucial molecules that play key roles in many biological processes (Figure 4, Figure 5 and Figure 6, Table 1). Some of these were equally distributed among the species or were found in two of the three; others were species-specific metabolites and this is reflected by the network analysis (Figure 7 and Appendix A, Table 1).

Vitamins not only are important co-factor of various enzymes that function in energy metabolism but they also play a key impact on immune responses [58]. Vitamin B2, enriched in cow MEVs, is well known to be an important co-factor of various enzymes that function in energy metabolism. It has been reported that this vitamin, in complex with a non-classical MHC class I molecule (MR1), stimulates innate mucosal-associated invariant T (MAIT) cells, which play an important role in mucosal defense and inflammation in the gut by producing IFN-γ and IL-17 [59]. Vitamin B9 (folate), enriched in goat MEVs, is also known to be essential for the maintenance of Regulatory T cells (Tregs) [60] and it has been reported that its deficiency causes intestinal inflammation [61].

“Vitamin B3 (nicotinate and nicotinamide) metabolism” is another MEV characteristic pathway. Indeed, in each species, metabolites such as NAD and its precursor Nicotinamide ribotide (NMN), NADH and NADP have been found highly abundant in P2 compared to the other pools. These molecules participate in catabolic redox reactions and are cofactors in anabolic redox reactions. Vitamin B3, also known as niacin, is a pharmacotherapeutic agent, firstly used in Pellagra treatment, highlighting its role in inflammation [62]. It has also been demonstrated that ligands of Hydroxycarboxylic acid receptors (HCA - GPR109A and GPR109B), where niacin is one of the most important, modulate lipopolysaccharide (LPS) mediated pro-inflammatory gene expression in both human macrophages and adipocytes and are beneficial in treating inflammation conditions [63], even the intestinal ones since these receptors are expressed by many cell types including immune cells and intestinal epithelium [64,65]. These benefices in intestinal inflammatory conditions such as inflammatory bowel diseases (IBD) have been demonstrated by Singh et al. [66] that showed the anti-inflammatory properties of GPR109A signaling in colonic macrophages and dendritic cells, enabling them to induce differentiation of Treg cells and IL-10-producing T cells. In this way, pharmacological treatment with niacin improves colitis conditions in a GPR109A-dependent manner [67].

Moreover, beyond the anti-inflammatory effects, Vitamin B3 is also able to decrease plasma triglycerides and low-density lipoprotein (LDL) cholesterol and increase high-density lipoprotein (HDL) cholesterol, improving cardiovascular diseases [68,69,70].

Among the amino acids, arginine (Arg) and tryptophan (Trp), enriched in goat MEVs, have been reported to have a large impact on the immune system [59]. Trp, a precursor of niacin, is an essential amino acid whose metabolism appears as a key modulator of gut microbiota and it has been hypothesized to play a role in the IBD syndrome, inducing Treg differentiation through microbiota-mediated degradation and kynurenine pathway (KP) [71]. In fact, in homeostasis conditions, Trp is converted into two molecules, Kynurenine (Kyn) by the host indoleamine 2,3-dioxygenase (IDO) and indole metabolites by gut microbiota [72]. Indole stimulates the mucosal defense through mucin production and tight junction protein enhancement. Moreover, Trp treatment in animal models of IBD such as DSS-induced colitis attenuates the symptoms and severity and decreases the expression of pro-inflammatory cytokines by the transcription factor aryl hydrocarbon receptor (Ahr) activation involved in the regulation of inflammation and immunity [73,74,75].

Arg is a conditionally essential amino acid in humans depending on the developmental stage and health status of the individual and has relevant nutrition properties. Its strong immune-potentiating effects are known [76]. Indeed, Arg is able to improve lipid profiles, inducing a decreasing serum concentration of triglycerides and in some cases of total cholesterol and LDL-cholesterol and also appears to increase antioxidant capacity while reducing oxidative stress and inflammation [77]. Moreover, it is well known that diet supplementation of l-arginine is effective in increasing plasma Arg that is a potential substrate for the enzyme endothelial nitric oxide synthase (eNOS), allowing the production of the vasodilator nitric oxide (NO) and leading to decrease blood pressure. This circulating Arg, in addition, induces eNOS transcription and downregulates hepatic key genes for fatty acid production in the liver [77,78]. Because of its metabolization to NO that has bactericidal properties, assisting macrophages and leukocytes in destroying microbial pathogens, Arg is considered an immunonutrient—it was observed indeed a decrease in lymphocyte T proliferation and NK-cell function by micro-environmental Arg starvation [59]. Moreover, it has been tested as an adjuvant in mice immunized with an inactivated vaccine, inducing total protection and antibody titers much higher than those receiving no Arg supplementation [79]. Arg probably plays a critical role in IBD patients which have been shown to have a decreased level of arginine due to the alteration of its metabolism in gut tissue, suggesting that dietary arginine supplementation might compensate for this deficiency in IBD tissues [80,81,82]. Moreover, Arg supplementation modulates the expression of both pro-inflammatory and anti-inflammatory cytokines especially increasing IL-10 that has an impact on secretory Immunoglobulin A (sIgA) control and therefore on intestinal mucosa protection [83]. Arg improves permeability and tight junctions (TJs) protein expression acting on the NF-κB signaling pathway and its related upregulation of anti-inflammatory cytokines, especially when integrated with the amino acid glutamine [84,85].

A similar effect on the intestinal barrier is induced by asparagine (Asn), another amino acid found in donkey MEVs, which is a precursor for many other amino acids such as aspartate, glutamine and glutamate. It has been demonstrated that Asn supplementation alleviates bacterial LPS-induced intestinal injuries, improving mucosa intestinal morphology and energy likely modulating the AMPK signaling pathway or inducing a decrease of intestinal pro-inflammatory cytokines and of enterocyte apoptosis or even regulating the CRF/CRFR1 signaling pathway, thus inducing inhibition of mast cell activation that arises as a response to this stress [86,87,88]. A possible role of Asn in immunomodulation has been demonstrated by Torres et al. after deprivation of this amino acid that gave rise to inhibition of T cell activation, characterized by the suppression of Myc expression and L-lactate secretion [88]. At the same time, a lack of Asn caused inhibition of activation-induced autophagy, essential for effector T cell survival and memory formation suggesting that both these processes require exogenous Asn [89]. In donkey MEVs also glutathione (GSH) was found; among its functions such as reserve form of cysteine and nitric oxide storage, the mainly known is its antioxidant capacity, either directly by interacting with reactive oxygen/nitrogen species or being oxidized by peroxidase enzymes in glutathione disulfide (GSSG) [90]. Glutathione deficiency is difficult to overcome through its oral administration [91] and MEVs could be a promising tool for improving the bioavailability of GSH containing endogenous peptides and acting directly as natural protection against the denaturing condition of the gastrointestinal tract.

Another potent immunomodulator found in goat MEVs is methylthioadenosine (MTA), a lipophilic sulfur-containing adenine nucleoside, which exerts specific pharmacological effects on cellular function [92]. It has been demonstrated that MTA is able to modulate the immune response suppressing the expression of pro-inflammatory cytokines like tumor necrosis factor-alpha (TNF-alpha) and enhancing those of interleukin-10 (IL-10) [93]. MTA markedly inhibits brain inflammation and autoimmune attack in the animal model of multiple sclerosis [94,95] and have been evaluated also in models of intestinal inflammation. MTA oral supplementation indeed prevented inflammation and tissue injury associated with DSS colitis in mice and reduced inflammation-induced colon cancer [96,97].

The anti-inflammatory role of MEVs is also strengthened by the presence of lysine in a consistent manner in the bovine milk and which is able to influence the inflammatory response through the modulation of pro and anti-inflammatory cytokines [98,99].

All these metabolites, enriched in MEVs take part in the “Urea cycle and metabolism of arginine, proline, glutamate, aspartate and asparagine” pathway, that emerged for all three species from network analysis (Figure 1c and Figure 7, Table 1). Furthermore, this pathway is shown in MSEA analyses (Figure 8) where we consider the entire content of metabolites for each type of MEVs (whole P2—Figure 1d), emphasizing the potential capacity of the cargo to become part of metabolic processes that can significantly influence the inflammatory response in the recipient cells.

Other MEV enriched amino acids have important roles in the maintenance of intestinal mucosa homeostasis. In particular, “methionine and cysteine metabolism” is a pathway that emerged from CREGN analysis in donkey and goat MEVs (Appendix A and Figure 1c, Table 1) and found also in bovine MEVs when considering the P2 whole cargo in MSEA analyses (Figure 8 and Figure 1d). Methionine (Met) and consequently its derivate cysteine, indeed, increase growth performance in piglets improving the integrity and the function of the small-intestinal mucosa while in mice and in poultry enhance the immune response in IBD reducing the susceptibility to colitis and improving inflammation and tissue injury [100]. Moreover Met has been used as mucosal protective during pharmacological treatment with cisplatin through antioxidative and anti-inflammatory effects and by enhancing the growth of beneficial bacteria [101]. In spite of these beneficial effects, recent studies on Met dietary-restriction have been shown the stimulation of the glutathione production and the improvement of the oxidative stress conditions in various organs, while the accumulation of Met leads to oxidative stress and histological changes [102]. Probably, Met plays a critical role in the intestinal imbalance mediated to oxidative stress and immunomodulation and in the host-microbiota interaction, even if these mechanisms need further investigations.

As regards the metabolites that differ between species and therefore belong to specific species pathways, surely goat MEVs were the most interesting both from a quantitative and qualitative point of view. For example, phenylalanine and tyrosine, as well as tryptophan, are aromatic amino acid ligands of Ca^2+^-sensing receptor (CaSR) that play a key role in the maintenance of intestinal barrier function [103]. Dietary supplementation with these amino acids alleviates piglet histopathological injury induced by LPS, decreases serum pro-inflammatory cytokines and significantly increases CaSR protein expression levels [84,103]. Choline, Arginine and Methionine, exclusive goat MEV metabolites, take part in “Glycine, serine, alanine and threonine metabolism” that emerges from the CREGN analysis (Table 1, Appendix A and Figure 1c) and is reported capable to protect intestinal mucosa during gut inflammation. Threonine and Serine indeed are necessary for intestinal mucosal protein synthesis, especially mucin, and for intestinal integrity, immune barrier function and oxidative status, also limiting the expansion of mucus degrading bacteria [104,105]. Glycine modulates the expression of pro-inflammatory and anti-inflammatory genes, improving the energy status and protein synthesis by regulating AMPK and mTOR signaling pathways and relieves inflammation by inhibition of TLR4 and NOD signaling pathways in LPS-challenged piglets [106].

The last main typology of MEV metabolites is some nucleotides and nucleosides, biological molecules which, in addition to fulfilling their main function as constituents of nucleic acids DNA and RNA, participate in a large part of the biological processes in the body. Indeed, pathways such as “Purine metabolism” and “Pyrimidine metabolism” were enriched in MEVs in both CREGN and MSEA analysis (Appendix A, Table 1) and for each species. Exogenous nucleotides supply may confer distinct biological benefits to tissues characterized by increased demand for nucleic acid synthesis, including gut injury, periods of rapid growth, immunosuppression, particularly those of the gastrointestinal and immune systems. Indeed, these metabolites accelerate recovery of the intestinal mucosa, improve barrier function, have strong anti-inflammatory capabilities and have the potential to reverse the oxidative stress secondary to inflammation and immunosuppression. Moreover, an influence on microbiota increasing the growth of beneficial bacteria and the potentiation of the immune response to pathogens has been demonstrated [107].

## 5. Conclusions

The MEV metabolites characterization highlights the presence of key molecules for immune and inflammatory response regulation. As emerged from the MSEA analysis of the entire MEVs cargo “aminoacyl-tRNA biosynthesis” is one of the most enriched pathways for all the species, demonstrating that the amino acid biosynthesis is the main activity carried by MEVs. At the same time, pathways for B vitamins and nucleotides were found enriched in CREGN analyses.

Moreover, CREGN analyses showed the presence of components that might be able to make potential beneficial modifications in the recipient cell at the inflammatory level. These MEVs metabolites, indeed, modulate the production of pro-inflammatory and anti-inflammatory cytokines, interact with receptors and other essential molecules for the activation of immune reactions and act on essential molecules for the maintenance of the microbiota and intestinal barrier functions and integrity. Immunomodulation and beneficial effects of these metabolites on the intestinal barrier have already been demonstrated, by improving conditions of chronic intestinal inflammation such as in IBD.

This peculiar message brought by enriched MEV metabolites and highlighted by CREGN analyses is often confirmed and strengthened by MSEA analysis of entire MEV cargos demonstrating these exclusive beneficial effects of the vesicles compared to the other milk components. In particular, the goat seems to be the species with the largest number of metabolites participating in these crucial pathways.

Our results, confirming the enriched presence of numerous key molecules, can be considered prodromal to test anti-inflammatory and immunomodulating effects of MEVs in vitro and in vivo on models of inflammatory-based diseases to evaluate their real efficacy and hypothesize a possible nutraceutical therapeutic approach.

## Figures and Tables

**Figure 1 nutrients-12-02908-f001:**
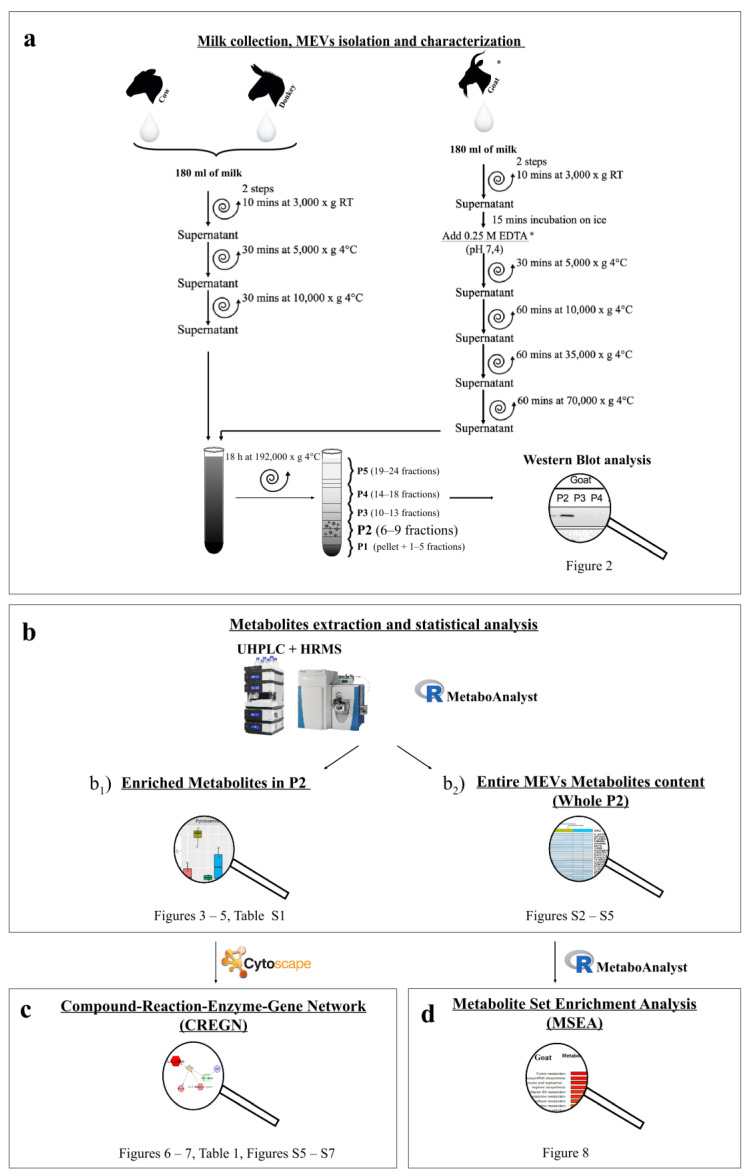
Workflow illustration and related scheme of results placement. (**a**) milk collection, milk extracellular vesicles (MEVs) isolation and characterization; (**b**) Metabolite extraction and statistical analysis; (**b_1_**) differentially abundant metabolites in P2 compared to the other pools; (**b_2_**) total metabolite content of MEVs in whole P2; (**c**) network analysis of enriched MEV metabolites; (**d**) pathway enrichment of entire MEV metabolites content. * EDTA: ethylenediaminetetraacetic acid tetrasodium salt dihydrate.

**Figure 2 nutrients-12-02908-f002:**
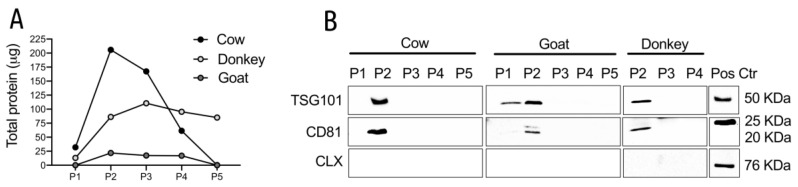
(**A**) Total amount of proteins in the pools of the gradient fractions from cow, goat and donkey milk. Data from representative experiments are reported. (**B**) Detection of the EV markers in the sucrose gradient pools of fractions. The indicated pools were immunoblotted with TSG101 (upper panels), CD81 (middle panels) and calnexin (lower panels). Each sample contained 30 μg proteins except for the donkey samples that contained 10 μg. The P1 and P5 pools from donkey milk were not tested because the total protein content was not sufficient. Positive controls were: Jurkat cells for CD81 and calnexin (CLX); cow MEVs isolated in previous experiments for TSG101. For each species at least two different preparations were tested; representative images are reported in the panels.

**Figure 3 nutrients-12-02908-f003:**
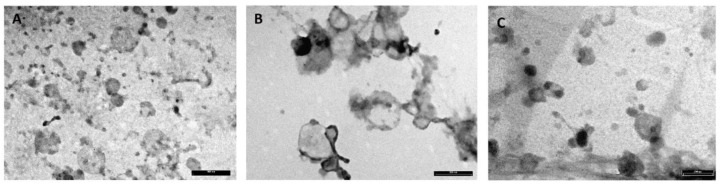
Electron micrograph showing MEVs isolated from donkey (**A**), bovine (**B**) and goat (**C**) milk. MEVs are round in shape and measure between 30 and 500 nm. Transmission electron microscopy (TEM), scale bar = 500 nm (**A**,**B**), 200 nm (**C**).

**Figure 4 nutrients-12-02908-f004:**
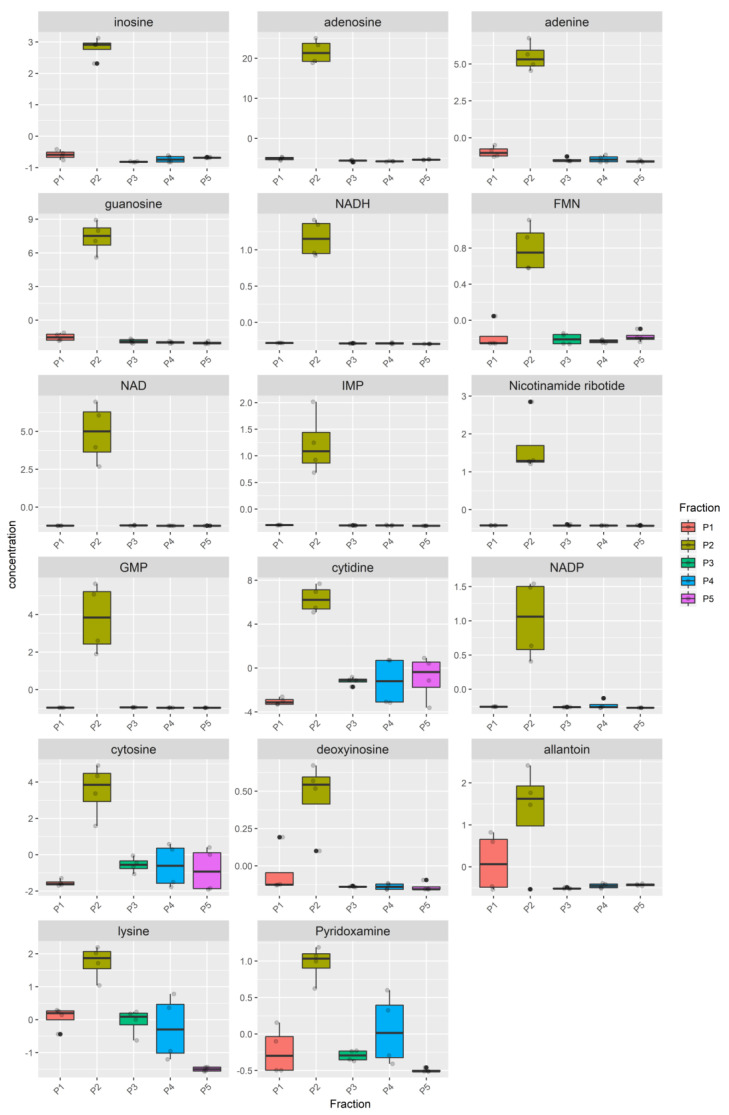
Metabolites enriched in the Bovine P2 fraction. For each fraction, four samples were analyzed. Raw data (quantitative relative ion count) were normalized using the Pareto Scaling normalization and analyzed using the ANOVA test. Only metabolites enriched in the Bovine P2 fraction are reported (FDR *p* < 0.05). The normalized concentration is reported for each metabolite in the P1, P2, P3, P4, P5 pools. The distribution of the samples is represented by a boxplot. The solid line represents the median, the box reports the interquartile range and the whiskers reports the minimum and maximum value of the distribution. Each sample is represented by a grey point, outliers are reported using a black point.

**Figure 5 nutrients-12-02908-f005:**
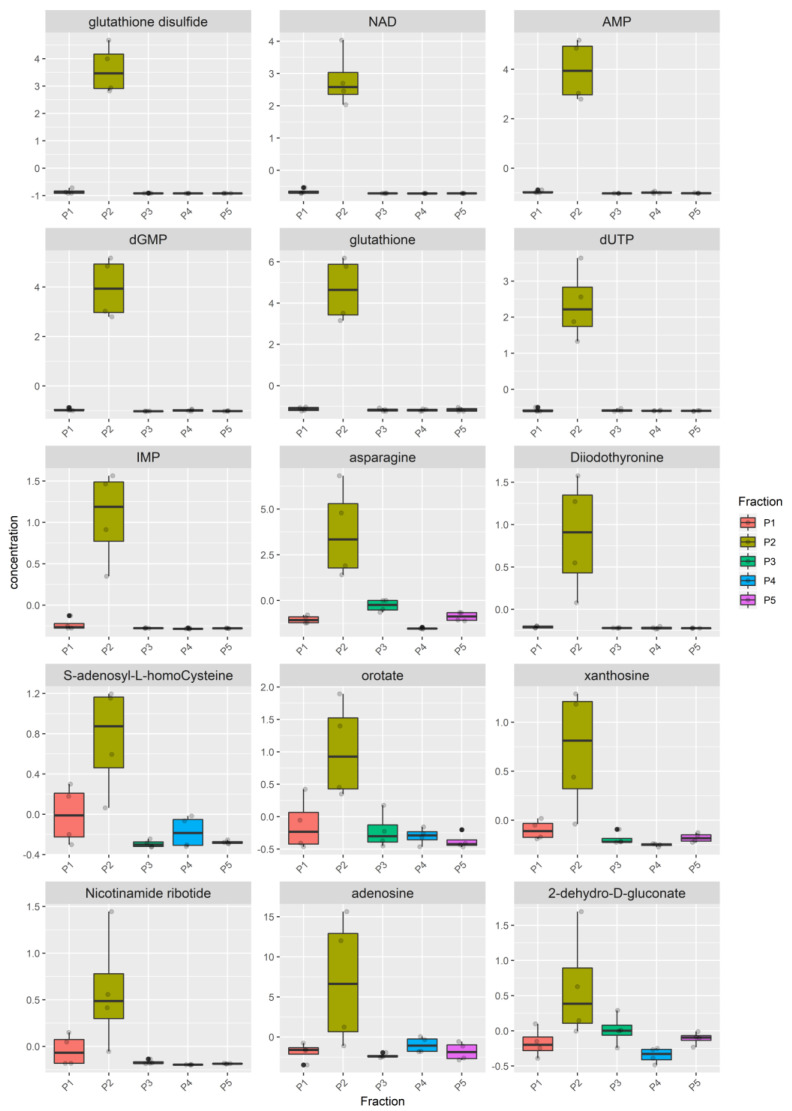
Metabolites enriched in the Donkey P2 fraction. For each fraction, four samples were analyzed. Raw data (quantitative relative ion count) were normalized using the Pareto Scaling normalization and analyzed using the ANOVA test. Only metabolites enriched in the Bovine P2 fraction are reported (FDR *p* < 0.05). The normalized concentration is reported for each metabolite in the P1, P2, P3, P4, P5 pools. The distribution of the samples is represented by a boxplot. The distribution of the samples is represented by a boxplot. The solid line represents the median, the box reports the interquartile range and the whiskers reports the minimum and maximum value of the distribution. Each sample is represented by a grey point, outliers are reported using a black point.

**Figure 6 nutrients-12-02908-f006:**
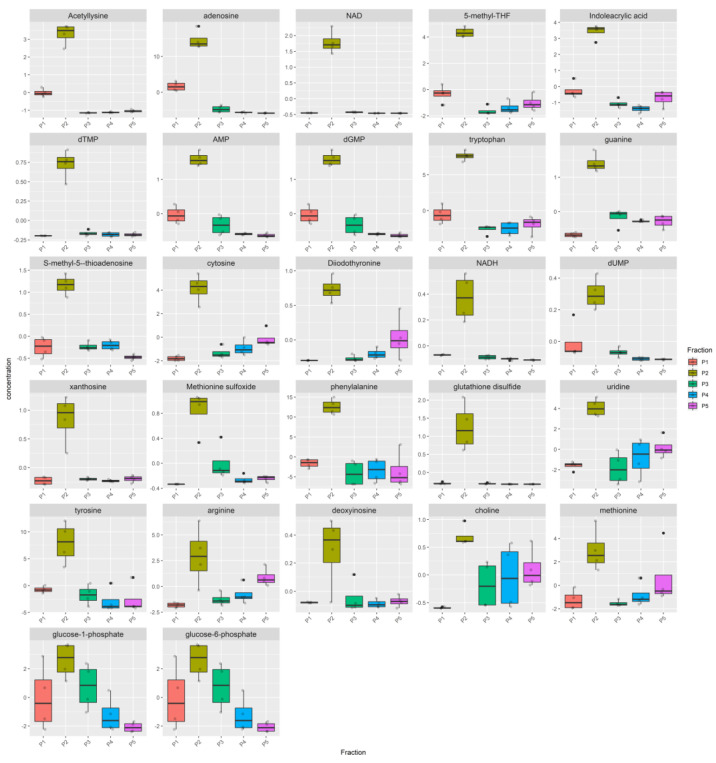
Metabolites enriched in the Goat P2 pool. For each fraction, four samples were analyzed. Raw data (quantitative relative ion count) were normalized using the Pareto Scaling normalization and analyzed using the ANOVA test. Only metabolites enriched in the Bovine P2 fraction are reported (FDR *p* < 0.05). The normalized concentration is reported for each metabolite in the P1, P2, P3, P4, P5 pools. The solid line represents the median, the box reports the interquartile range and the whiskers reports the minimum and maximum value of the distribution. Each sample is represented by a grey point, outliers are reported using a black point.

**Figure 7 nutrients-12-02908-f007:**
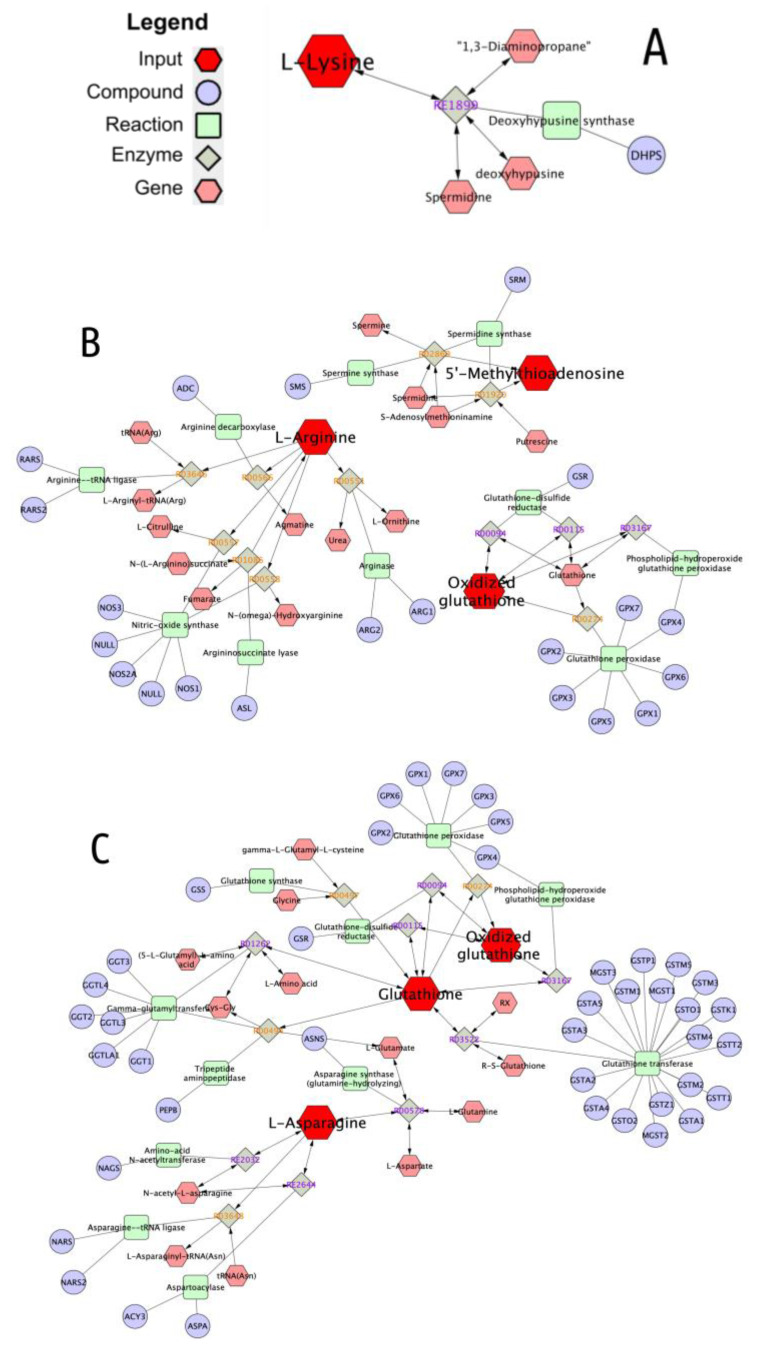
Visual representation generated by MetScape (CREGN) application of the Urea cycle and metabolism of arginine, proline, glutamate, aspartate and asparagine, a common pathway among the three species. Red hexagons symbolize these input MEV metabolites, arrows indicate their connections with other intermediate compounds (pink hexagons) and enzymes (green squares), which might regulate the identified metabolites. Purple circles represent genes encoding those enzymes and beige rhombuses the reactions catalyzed by those enzymes. (**A**) Bovine, (**B**) Donkey, (**C**) Goat.

**Figure 8 nutrients-12-02908-f008:**
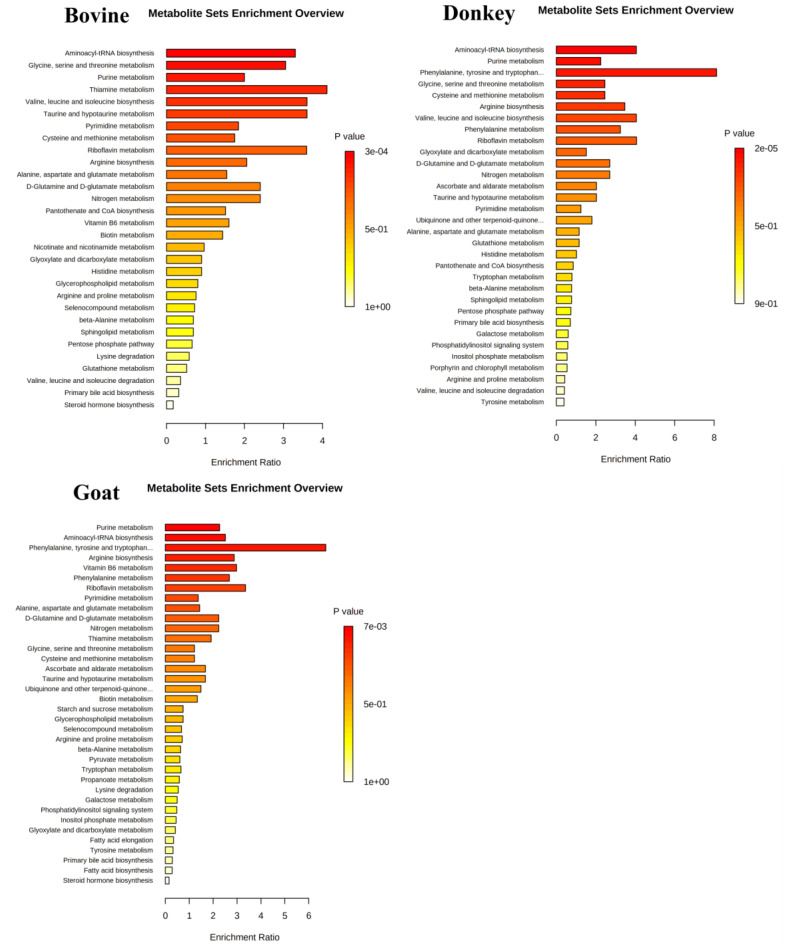
Metabolite Set Enrichment Analysis (MSEA) overview obtained through MetaboAnalyst by plotting -log of *p*-values from pathway enrichment analysis on the *y*-axis and pathway impact values from pathway topology analysis on the *x*-axis. Color intensity (light yellow to red) reflects increasing statistical significance.

**Table 1 nutrients-12-02908-t001:** Pathways and related MEV metabolites summary.

Pathways	Pathway Recurrence among Species	Bovine Metabolites	Donkey Metabolites	Goat Metabolites
Purine metabolism	3	Adenine Adenosine; Deoxyinosine; GMP; IMP; Inosine; Guanosine	Adenosine; AMP; dGMP; IMP; Xanthosine	Adenosine; AMP; Deoxyinosine; dGMP; Guanine; Xanthosine
Pyrimidine metabolism	3	Cytidine	dUTP Orotate	dTMP; dUMP; Uridine
Urea cycle and metabolism of arginine, proline, glutamate, aspartate and asparagine	3	Lysine	Glutathione; Glutathione disulfide; Asparagine	S-methyl-5-thioadenosine; Glutathione disulfide; Arginine
Vitamin B3 (nicotinate and nicotinamide) metabolism	3	NAD; NADH; NADP; Nicotinamide ribotide	NAD; Nicotinamide ribotide	Arginine; NAD; NADH
Methionine and cysteine metabolism	2		S-Adenosyl-L-homocysteine	Methionine
Tyrosine metabolism	2		Diiodothyronine	Diiodothyronine; Phenylalanine; Tyrosine
Pentose phosphate pathway	1		2-Dehydro-D-gluconate	
Biopterin metabolism	1			Phenylalanine; Tyrosine
Galactose metabolism	1			Glucose-1-phosphate
Glycerophospholipid metabolism	1			Choline
Glycine, serine, alanine and threonine metabolism	1			Choline; Arginine; Methionine
Glycolysis and Gluconeogenesis	1			Glucose-1-phosphate
Tryptophan metabolism	1			Tryptophan
Vitamin B9 (folate) metabolism	1			5-methyl-THF
Vitamin B2 (riboflavin) metabolism	1	FMN		
Vitamin B6 (pyridoxine) metabolism	1	Pyridoxamine		
Vitamin H (biotin) metabolism	1	Lysine		
Lipoate metabolism	1	Lysine		
Lysine metabolism	1	Lysine		
None ^1^		Allantoin; Cytosine		Acetyllysine; Cytosine; Deoxyribose-phosphate Glucose-6-phosphate Indoleacrylic acid
No KEGG ID ^2^				Methionine sulfoxide; 5-deoxyribose-1-phosphate; N(alpha)-Acetyllysine

^1^ No pathways were found for the reported metabolites; ^2^ No KEGG ID are associated to the reported metabolites.

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
