# Peer review of "Anti-Inflammatory Potential of Cow, Donkey and Goat Milk Extracellular Vesicles as Revealed by Metabolomic Profile"

_nutrients, 2020, doi:10.3390/nu12102908_

Round 1

Reviewer 1 Report

This is an interesting topic and worth analytical effort. However, the data and their presentation is partially hard to digest and the conclusions partly not "round and sound". The paper must be written more condensed and focussed, the methods more detailed.

Major:

Why was no comparative analysis made on human milk as a reference, using the same procedures? The authors should provide a scientific reasoning of the chosen species, as donkey for milk does not play a major role in other countries of the EU. Why not horse and sheep as well? The general differences between human, ruminant and equine milk (macronutrients, glycoconjugates etc) should be pointed out.

The authors provide no quantitative information. There is no data how much P2 in relation to P1+P3-P5 exists. There is no information whether the amounts of components in P2 are quantitativey relevant in compariso to whole milks.

Fig 1: Redraw and show only links to Figures. b1,b2, c and d are much too small.

Line 123 ff: How many different samples of milk from the animal species were analysed? As it is written, there is one mixed sample from the three animal species. Hence, there is no justification for mean, standard deviations, ranges and statistics as outlined in figure below. There is no consistent information about varability within a species. Is the feeding of the animals generally "physiologic", particularly of the cows in comparison to the goats?

line 131ff: What does EDTA and other differences in preparation to the process? EDTA effects must be tested for the other milk species.

line 157ff: I'd like to see electron microscopy pics of the preparations.

line 173ff: The mass spec analysis is not sufficiently described in terms of molecular identity/diagnostic fragments of compounds.
The authors provide no information about the lipids of the MSVs, although there must be plenty of them. They are all below 1kDa and must have been present in the solvent mixture and be detectably in the scans.

Line 239: Which statistical tests? They must be described in Materials and Methods.

Fig. 3-5: Create figures instead, where all components are in the same order and show it in comparison for the three animal species. This will better visualize the differences between cow, goat and donkey.
Explain the bars and boxes. Where do the standard deviations/ranges come from. Indicate numbers of samples (not analysis replications!) and the way of presentation (means, medians, SD or SE and so on).

Fig. 6 and 7: These figures are overloaded and unreadable due to small fonds. There is no adequate legend to explain, what the message is. What is the meaning of such untargeted drawings? E.g., does the pentose phosphate pathway only apply to the donkey? To the opinion of the reviewer this is all confusing and spoils the key messages of the relevance of MSVs!

Fig. 8/line 322: The diction pathway enrichment analysis is weird in relation to the compounds found in the P2. Or did the authors measure the enzymes in their preparations?

Discussion: The general question is the advantage of MSVs compared to the free substances. The examples and references shown provide no conclusive evidence for this, and for the preparation of MSVs for therapy/prevention, with the exception of glutathione.

To the reviewer's opinion the argumentation must focuss a completely different aspect: why does the newborn/infant/calf etc. require niacin, purines and pyrimidines in the form of nucleotides? Is it better for growth? Might it be advantageous in diseases requiring intense repair of enterocytes? Are the compounds tranferred to the circulation as intact molecules, possibly exonerating the liver and other organs from energy-consuming synthesis of NAD(P)H from niacin? What individual compounds are of reasonable impact for e.g. chronic inflammatory bowel diseases?

Minor:

Line 36f: Diction. How can particles and metabolites below 1kDa contain pathways? Were the enzymes in them measured as well?

Line 74: write "colostrum and >mature< milk".

Line 80: >taken up< rather than >uptaked<. This word does not exist.

Fig S1: should be in the main manuscript,

Fig S2ff: The figures are poorly designed/resolved. The fonds are too small

line 169: How long were samples stored at -20°C?

line 248: ... 15 show..., rather than >shows<.

line 288f: What do the authors mean by this:  ... as valine, leucine and isoleucine biosynthesis were present in bovine and donkey... These are all essential amino acids.

line 326: remove second full stop.

Table 1: Tyrosine metabolism. The diiodo. thyrosine is more related to iodine and thyroid gland metabolism rather than thyrosine.

line 348: What do the authors mean by by "peculiar" and by "message"?

Reviewer 2 Report

The manuscript proposed by Mecocci S L et al. about the ‘Anti-inflammatory Potential of Cow, Donkey and Goat Milk Extracellular Vesicles As Revealed By Metabolomic Profile’.

The authors performed a metabolomic analysis of MEVs, from bovine, goat and donkey, by mass spectroscopy (MS) coupled with Ultrahigh-performance liquid chromatography (UHPLC). Metabolites were identified, and enriched metabolic pathways were investigated; the authors reported that the final aim was to evaluate their anti-inflammatory and immunomodulatory properties in view of nutraceutical applications in inflammatory conditions.

Notably, as EVs share several physicochemical properties with other nanostructures present in milk, the authors performed separation on a sucrose gradient to overcome to overcome limits and non-specific data.

The manuscript shows preliminary results (exclusively isolation and characterizaion of MEVs and metabolomic analysis), is overall well written and requires minor revisions.

Figure 1 shows a workflow illustration and related scheme of results placement; even if Figure 2 is shown after, the authors could improve the resolution of western blotting analysis that is not clearly seen here.

The authors correclty isolated MEVs; anyway, molecular characterization alone is insufficient: the authors need to introduce a dimensional chracterization of MEVs such as TEM or Nanoparticle tracking analysis (NTA).

Supplementary Figure S2-S3-S4: not readable, improve the resolution.

Round 2

Reviewer 1 Report

The authors have answered many queries, and clearly pointed out the reason, why these species were selected. The arguments, why sheep were excluded, is not convincing for other parts of the EU.

They have partly refused suggestions concerning the biological meaning of MEVs for developing organisms (which might be a basis for future applications). Neverteless, the manuscript has significantly been improved!

There is still a formal problem with the font size  in figures and table 1. This problem cannot be solved by higher magnification on the screen, due to the resolution of the original. This equally applies to the supplement.

The supplementary tables, as indicated in the response and main text, do not exist in the provided supplement. At least they were not to be downloaded. The authors should complete this!

The mix-up in the authors' formulation on the presence of pathways in MEVs has not consistently been solved (Disc., l. 429). Moreover (l429), metabolism is used synonymous with pathway here.

Minor:

l 431: compared to, not with respect to.

l. 436: lipopolysaccharide, no capital

l. 469 >to< rather than >in<, comma after properties.

l. 475: decreased, rather than decreasing.

l. 476: delete >the<.

l. 493, 496: glutathione without capital.

l. 498: introduce abbreviation for GSH

l. 500 metahylthioadenosine without capital.

l. 519: methionine without capital; "metabolism" is not a "pathway"!

l. 565: aminoacyl- without capital.
